# Effect of Atomic Size Difference on the Microstructure and Mechanical Properties of High-Entropy Alloys

**DOI:** 10.3390/e20120967

**Published:** 2018-12-14

**Authors:** Chan-Sheng Wu, Ping-Hsiu Tsai, Chia-Ming Kuo, Che-Wei Tsai

**Affiliations:** 1Department of Materials Science and Engineering, National Tsing Hua University, Hsinchu 30013, Taiwan; 2High Entropy Materials Center, National Tsing Hua University, Hsinchu 30013, Taiwan

**Keywords:** high-entropy alloys, mechanical property, recrystallization

## Abstract

The effects of atomic size difference on the microstructure and mechanical properties of single face-centered cubic (FCC) phase high-entropy alloys are studied. Single FCC phase high-entropy alloys, namely, CoCrFeMnNi, Al_0.2_CoCrFeMnNi, and Al_0.3_CoCrCu_0.3_FeNi, display good workability. The recrystallization and grain growth rates are compared during annealing. Adding Al with 0.2 molar ratio into CoCrFeMnNi retains the single FCC phase. Its atomic size difference increases from 1.18% to 2.77%, and the activation energy of grain growth becomes larger than that of CoCrFeMnNi. The as-homogenized state of Al_0.3_CoCrCu_0.3_FeNi high-entropy alloy becomes a single FCC structure. Its atomic size difference is 3.65%, and the grain growth activation energy is the largest among these three kinds of single-phase high-entropy alloys. At ambient temperature, the mechanical properties of Al_0.3_CoCrCu_0.3_FeNi are better than those of CoCrFeMnNi because of high lattice distortion and high solid solution hardening.

## 1. Introduction

A multi-principal-element alloying system, developed by Yeh et al. in 2004 [1,2], is called high-entropy alloy (HEA). HEAs are defined as equiatomic or near-equiatomic alloys containing at least five elements, whose atomic concentration ranges from 5% to 35%. The multiprincipal-elemental mixtures of HEAs result in high entropy, lattice distortion, sluggish diffusion, and cocktail effects [3]. High entropy causes single-phase structures to become stable; as such, HEAs usually consist of simple solid solution phases with face-centered cubic (FCC) and body-centered cubic (BCC) structures rather than other intermetallic compounds [2]. A lattice is highly distorted because all atoms are solutes that can disorderly fill in a FCC or BCC lattice, and atomic sizes differ among elements in this alloying system, thereby possessing strengthening effect in high-solid-soluted HEAs [4]. Lattice distortion impedes atomic movement and slows down the diffusion rate of atoms in HEAs; these conditions lead to higher recrystallization temperatures, that is, the activation energies of grain growth in deformed HEAs are higher than those in conventional alloys [1,5].

In FCC single-phase HEAs, CoCrFeMnNi has been widely studied [2,6,7,8,9,10,11]. The atomic radius of each component in CoCrFeMnNi is close to each other, and the atomic size difference (δ) is approximately 1.18% only. δ is defined as follows:(1)$$\delta =100\sqrt{\sum_{i=1}^{n}c_{i}\;{\left(1-r_{i}/\bar{r}\right)}^{2}},$$
where $\bar{r}=\overset{n}{\underset{i=1}{\sum }}c_{i}r_{i}\;$ is the average radius, $c_{i}$ and $r_{i}$ are the atomic percentage and atomic radius of the i element, respectively [12,13,14]. Other reports have also suggested that δ should consider the shear modulus, adjacency, and differences in the modulus of another element in single BCC phase HEA [15]. Moreover, researchers clarified that δ might be dependent on chemical composition of the alloys and the local lattice distortion in HEAs would differ from place to place [16,17].

Grain growth kinetics is usually described by the following equation:(2)$$d^{n}-d_{0}^{n}=kt,$$
where *d* is the grain size after annealing time *t*, *d*_0_ is the initial grain size (this term approaches to zero for the as-rolled state), *k* is a temperature-dependent constant, and *n* is the grain growth exponent. *k* can be described by Equation (3):(3)$$k=k_{0}e^{-Q/RT},$$
where *k_0_* is a constant, *R* is Boltzmann’s constant, *T* is the temperature, and *Q* is the activation energy of grain growth.

Otto et al. [6] observed that the activation energy of the grain growth of CoCrFeMnNi is 325 kJ/mol. However, the relation between lattice distortion and grain growth activation energy remains unknown. To clarify this phenomenon, other researchers [9,18] added aluminum, whose atomic radius is larger than that of other elements, to CoCrFeMnNi. Al is a BCC stabilizer in HEAs [9,19]. Thus, Al addition is limited to a molar ratio of 0.2 to prevent the formation of dual-phases in CoCrFeMnNi. With a moderate amount of Al addition, a FCC single-phase structure and a high atomic size difference should be obtained in the designed alloy.

Al_0.5_CoCrCuFeNi high-entropy alloy is composed of matrix and Cu-rich phases, which become a FCC phase after they are treated with a solution [19,20,21,22,23]. A Ni–Al–rich phase precipitates in the matrix phase, and a Cu-rich phase precipitates at midtemperature, leading to the brittleness of the alloy at intermediate temperature. Thus, to design single-phase structures, decreasing the amounts of Al and Cu in Al_0.5_CoCrCuFeNi is needed.

According to the Hume–Rothery rule, solid solutions can be obtained until the atomic size difference is larger than 15% [24]. In addition, factor Ω can be utilized to predict whether an alloy is solid soluted:(4)$$\Omega =T_{m}\frac{\Delta S_{mix}}{\left|\Delta H_{mix}\right|},$$
where *T_m_* is the melting point (Kelvin) of the alloy, and Δ*S*_mix_ and Δ*H*_mix_ are the mixing entropy and enthalpy of the alloy, respectively [13]. For solid solution formation, Ω should be larger than 1. Δ*H*_mix_ ranging from −15 kJ/mol to 5 kJ/mol is beneficial to obtaining single-phase structures [12].

For high-entropy alloys, an empirical statistical pattern is observed by calculating the valence electron concentration (VEC) to predict the formation of stable phases [25,26]. A phase consists of BCC and FCC when the VEC is below 6.8 and above 8.0, respectively. Otherwise, BCC and FCC phases are formed in HEAs.

In this study, Al_0.3_CoCrCu_0.3_FeNi (δ = 3.65%; ΔH_mix_ = −4.38 kJ/mol; Ω = 5.67; VEC = 8.35) is designed to be the FCC single-phase high-entropy alloy. The lattice distortion and activation energy of grain growth in Al_0.3_CoCrCu_0.3_FeNi are compared with those of other FCC single-phase HEAs to determine the effect of atomic size difference on microstructure and mechanical properties.

## 2. Materials and Experimental Methods

High-entropy CoCrFeMnNi, Al_0.2_CoCrFeMnNi, and Al_0.3_CoCrCu_0.3_FeNi alloys were prepared by arc melting in a vacuum chamber at a pressure of 0.01 Torr, and its constituent elements had at least 99.99 wt.% purity. Melting was performed at a current of 500 A in a water-cooled copper hearth and repeated at least four times to confirm chemical homogeneity. The dimensions of the final solidified ingot were cuboid and had a width of 20 mm, a length of 40 mm, and a height of 10 mm. The ingots were homogenized at 1100 °C for 6 h, quenched with water quenching or cooled in a furnace, and cold rolled at a thickness reduction of 70%. The specimens were annealed between 900 °C and 1100 °C for various times and finally quenched with water.

The specimens were prepared by cutting, grinding, and polishing in a sequence. The crystalline structure of the present alloys was characterized using an X-ray diffractometer SHIMADZU-XRD6000 equipped with Cu-target radiation (K_α_ = 1.54 Å) at 30 kV and 20 mA. The sample was scanned at 2θ angle from 20° to 100° at a scanning rate of 2°/min. The microstructures were observed under a scanning electron microscope (JEOL-5410) at an acceleration voltage of 20 kV for a working distance of 24 mm. All of the specimens were etched with 0.5 g of copper (II) chloride, 10 mL of hydrochloric acid, and 10 mL of ethanol mixing liquid solution to observe the grains. The grain size was calculated and statistically measured using ImageJ. The grain size was also circled at least 400 grains to yield an average grain size for each specimen. For the tensile test, all of the specimens were tested at ambient temperature by Instron 4468 at a stain rate of 10^−3^ s^−1^. Figure 1 presents the dimensions of the samples used in the tensile test.

## 3. Results

### 3.1. Microstructure and Crystalline Structure

Al_0.2_CoCrFeMnNi is designed from CoCrFeMnNi by adding 0.2 molar ratio of Al to contribute increased lattice mismatch. Figure 2 shows the X-ray diffraction (XRD) analysis of Al_0.2_CoCrFeMnNi HEAs in its homogenized state quenched with water and cooled in a furnace. The lattice constant of Al_0.2_CoCrFeMnNi is 3.582 Å. The atomic size difference of Al_0.2_CoCrFeMnNi can be calculated by Equation (1), and the value is 2.77%. Figure 3 shows the XRD analysis of the designed Al_0.3_CoCrCu_0.3_FeNi HEAs in its homogenized state after water quenching and furnace cooling. The diffraction pattern also clearly reveals the appearance of peaks of the FCC structure only. The lattice constant of Al_0.3_CoCrCu_0.3_FeNi is 3.585 Å. The atomic size difference of Al_0.3_CoCrCu_0.3_FeNi is 3.65%, which is calculated by Equation (1).

### 3.2. Grain Growth Activation Energy of Al_0.2_CoCrFeMnNi and Al_0.3_CoCrCu_0.3_FeNi Alloys

All of the present alloys exhibit only single FCC phases without any other precipitations and show good workability at room temperature. The reduction of thickness can reach 70% without any cracks neither on the rims nor inside the as-rolled specimens. Recrystallization occurs after annealing and is performed above 900 °C for various times. The movement of each solute atom is more difficult than that in traditional alloying systems because of sluggish effect in high-entropy alloys. Under this condition, the recrystallization temperature in HEAs becomes higher than that in conventional alloys. Recrystallization takes place in high-lattice-strain energy regions, such as slip band, deformation twin intersections, and grain boundaries, which are the preferred nucleation sites for new strain-free grains. The average grain size of Al_0.2_CoCrFeMnNi is 42.6 μm after it is annealed at 1000 °C for 120 min, water quenching is subsequently performed Figure 4a. The grain sizes at different annealing temperatures for various times are shown in Table 1. The grain growth activation of Al_0.2_CoCrFeMnNi is calculated with Equations (2) and (3), and the linear fitting result is shown in Figure 5. The slope of the fitting line represents the activation energy (Q) for grain growth, which is 434.4 kJ/mol in Al_0.2_CoCrFeMnNi.

The grain growth is observed at 900 °C in Al_0.3_CoCrCu_0.3_FeNi, and the evolution of the microstructure in Al_0.3_CoCrCu_0.3_FeNi is shown in Figure 6. High-resolution scanning electron microscopy is utilized after the etching condition is optimized for observation because of the small grain size. The grain sizes at different temperatures in various times are shown in Table 2. The grain growth activation is calculated by Equations (2) and (3), and the fitting result is shown in Figure 7. The activation energy of grain growth is 761.3 kJ/mol.

### 3.3. Relationship between Atomic Size Difference and Grain Growth Activation Energy

Three high-entropy alloy systems with a single FCC phase are observed: CoCrFeMnNi, Al_0.2_CoCrFeMnNi, and Al_0.3_CoCrCu_0.3_FeNi. These alloys are selected and compared with others. The outstanding phase stability of these HEAs can avoid from the formation of second phases that can influence grain growth at increased temperature.

The calculated results of atomic size difference and grain growth activation is shown in Table 3. The atomic size difference of Al_0.3_CoCrCu_0.3_FeNi is 3.65% as calculated by Equation (1). This value is the highest in these three kinds of HEAs, and 2.77% and 1.18% belong to Al_0.2_CoCrFeMnNi and in CoCrFeMnNi, respectively. The activation energy of Al_0.3_CoCrCu_0.3_FeNi for grain growth is also the highest among others.

The mixing enthalpy between each solute element is also shown in Table 4 to determine how the mixing enthalpy affects the activation energy of grain growth. However, the mixing enthalpy of these three HEAs is slightly related to the atomic size difference or the activation energy of grain growth.

## 4. Discussion

### 4.1. Effect of Atomic Size Difference on Microstructures

The comparison result of the microstructure between Al_0.2_CoCrFeMnNi and Al_0.3_CoCrCu_0.3_FeNi single FCC-type high-entropy alloys according to XRD analysis in Figure 2 and Figure 3 shows that the as-homogenized state and the furnace-cooled state both have outstanding phase stability without any detrimental non-FCC phases regardless of the cooling condition. Haas, Sebastian, et al. [27] reported that the Gibbs free energy of solid solution alloys is completely due to configurational entropy and contributes to the thermal stability of solid solution alloys. The consequent single FCC phase is gained with a significant sluggish diffusion effect in HEAs [3], and this parameter is beneficial to the following heavily cold-rolling procedure without showing the undesired brittleness and ensuring great workability.

The atomic size difference of Al_0.3_CoCrCu_0.3_FeNi is larger than that of Al_0.2_CoCrFeMnNi. Different grain sizes are found (Table 2 and Table 3) when the samples are treated with the same thermomechanical process, that is, 70% cold rolled, annealed at 1000 °C for 120 min, and quenched with water. This result reveals that the atomic size difference likely affects the ability of dislocation movement, causing a different grain growth behavior in these two HEAs.

### 4.2. Effect of Atomic Size Difference on Mechanical Properties

Large atomic size difference (δ) corresponds to the great amount of activation energy needed for grain growth. Large δ is associated with a high degree of lattice distortion in the single-phase solid solution and cause atoms to spontaneously move at the most stable state, thereby decreasing the potential energy of the existing defects in a low level; that is, defects are found in stable sites. The energy of grain growth comes from the different energy levels between lattice distortion and defects [28]. When the number of alloying elements is low, or the alloy is low entropy, the energy of grain growth in such an alloy, which is nearly pure metal, is attributed to undistorted grain and perfect dislocation, and the energy level difference between the former and the latter can be regarded as the driving force for recrystallization then to make a perfect dislocation being released by a new grain.

In other cases, if the difference between these two energies is insufficient, the driving force for recrystallization is too low to induce dislocation rearrangement. In other words, higher annealing temperature or longer annealing time is needed so that it can enable the recrystallization to start at dislocation site and new grain to grow subsequently.

Table 4 shows that aluminum has the highest binding energy to each element, possibly leading to the solute-pinning effect and making each solute atom suitable to their lattice site. In other words, atoms become self-accommodating to the stable sites. Thus, when a great amount of aluminum is added, the grain is distorted remarkably because of large δ, causing energy level of a grain to be higher than undistorted one. Also, solute atoms are pinned in their preferred lattice site, and each solute atom can act as an obstacle of the movement of dislocations, thus obtaining the lower energy level of pinned dislocations, that is, dislocations become more stable or immovable.

### 4.3. Comparison of Tensile Properties with Different HEAs

Otto et al. [6] reported that the tensile properties of CoCrFeMnNi at different temperatures with different grain sizes are yield strength of 350 MPa, ultimate tensile strength of 650 MPa, and recrystallized grain size of 4.4 μm. With cold rolling and annealing, the grain size of CoCrFeMnNi can be small. The compatible small scale of the grain size of Al_0.3_CoCrCu_0.3_FeNi was designed to compare its mechanical properties with those of CoCrFeMnNi. After cold rolling and annealing were performed at 900 °C for 5 h and water quenching was conducted, the grain size is approximately 5.13 μm under the optimized thermomechanical treatment. The final microstructure is shown in Figure 6b.

Figure 8 illustrates the tensile stress–strain curves of the designed Al_0.3_CoCrCu_0.3_FeNi in the as-homogenized states and 900 °C/5 h annealing state. Table 5 shows the comparison of the mechanical properties between CoCrFeMnNi and Al_0.3_CoCrCu_0.3_FeNi with different grain sizes. The yield strength and the ultimate tensile strength are 500 and 717 MPa in the annealed Al_0.3_CoCrCu_0.3_FeNi, respectively. The elongation of Al_0.3_CoCrCu_0.3_FeNi is smaller than that of CoCrFeMnNi, suggesting that the plastic deformation in CoCrFeMnNi is involved in one-to-one atom-vacancy exchange mechanism [29,30]. This finding can be accounted for this high ductility.

The lattice distortion of Al_0.3_CoCrCu_0.3_FeNi is higher than that of CoCrFeMnNi because the former has a larger atomic size difference (δ) than the latter. Different δ can be related to the mechanical behavior, for an example, higher yield and tensile strength in Al_0.3_CoCrCu_0.3_FeNi with the same FCC structure and the close value of grain sizes compared with CoCrFeMnNi. The high lattice distortion can introduce the concentrated strain field around the lattice, causing the movement of dislocations to be more difficult. The large amount of the added aluminum can introduce a high degree of interaction to each alloying element, causing the pinning effect on dislocations. Finally, in high-entropy alloys with high lattice distortion, although the lattice distortion effect on properties of HEAs is yet an open question still await to be solved [31], once a large atomic size difference is obtained, the increment of tensile strength can be predicted.

## 5. Conclusions

Al_0.2_CoCrFeMnNi is a single FCC high-entropy alloy structure composed of Al added to CoCrFeMnNi at a molar ratio of up to 0.2. The microstructure of Al_0.3_CoCrCu_0.3_FeNi is also a single FCC phase. The recrystallization and grain growth behavior of Al_0.3_CoCrCu_0.3_FeNi and Al_0.2_CoCrFeMnNi are observed. The calculation of the grain sizes under different annealing conditions reveals that the activation energy of the grain growth of Al_0.2_CoCrFeMnNi is 434.4 kJ/mol with an atomic size difference of 2.77%. The activation energy of the grain growth of Al_0.3_CoCrCu_0.3_FeNi is 761.3 kJ/mol with an atomic size difference of 3.65%. A large atomic size difference indicates that a high activation energy is needed for grain growth. The lattice distortion of Al_0.3_CoCrCu_0.3_FeNi is much higher than that of single FCC-phase CoCrFeMnNi. The mechanical properties of Al_0.3_CoCrCu_0.3_FeNi are superior to those of CoCrFeMnNi under similar conditions. This result is attributed to high lattice distortion and pinning effect on dislocation because of the large atomic size difference in high-entropy Al_0.3_CoCrCu_0.3_FeNi alloy.

## Figures and Tables

**Figure 1 entropy-20-00967-f001:**
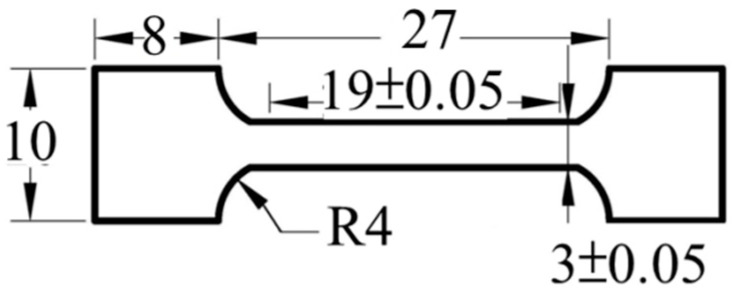
Dimensions of tensile specimens (unit: mm).

**Figure 2 entropy-20-00967-f002:**
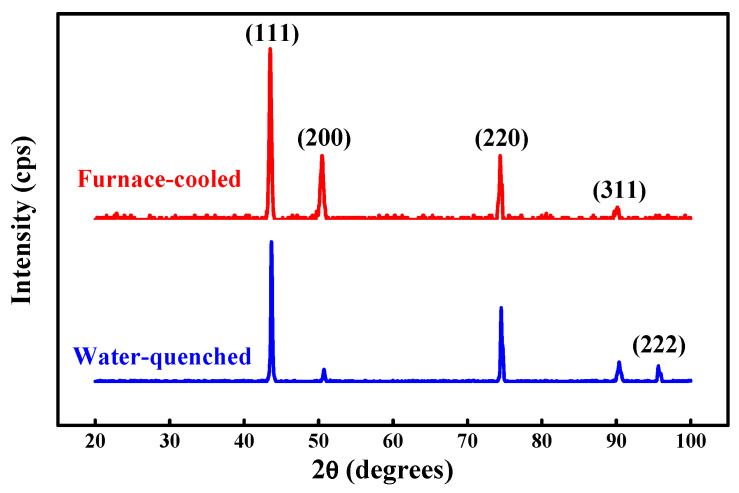
X-ray diffraction (XRD) patterns of homogenized Al_0.2_CoCrFeMnNi alloys after water quenching and furnace cooling.

**Figure 3 entropy-20-00967-f003:**
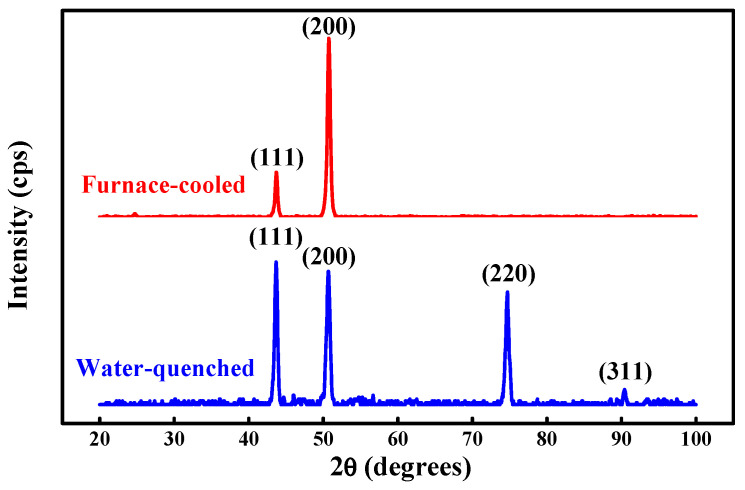
XRD patterns of homogenized Al_0.3_CoCrCu_0.3_FeNi alloys after water quenching and furnace cooling.

**Figure 4 entropy-20-00967-f004:**
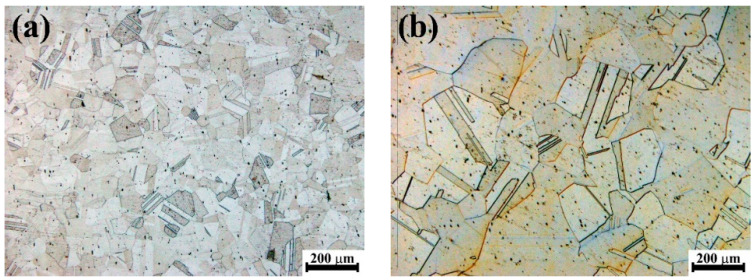
Microstructure of Al_0.2_CoCrFeMnNi with cold rolling and annealing at (**a**) 1000 °C and (**b**) 1100 °C for 120 min.

**Figure 5 entropy-20-00967-f005:**
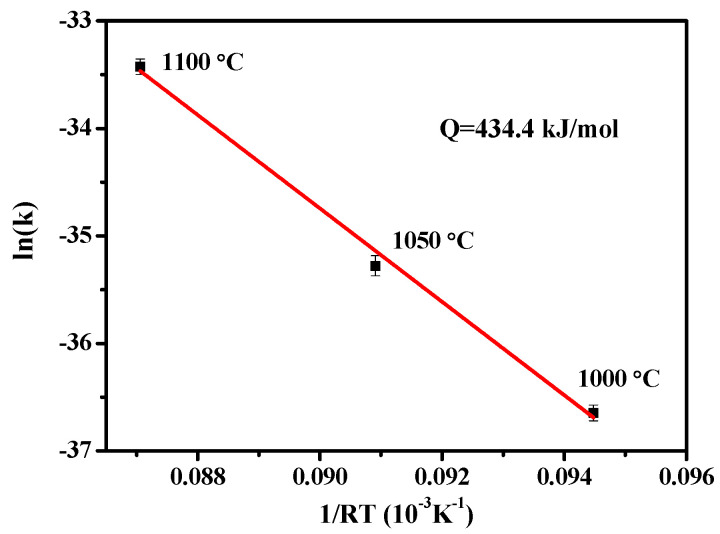
Grain growth activation energy of Al_0.2_CoCrFeMnNi.

**Figure 6 entropy-20-00967-f006:**
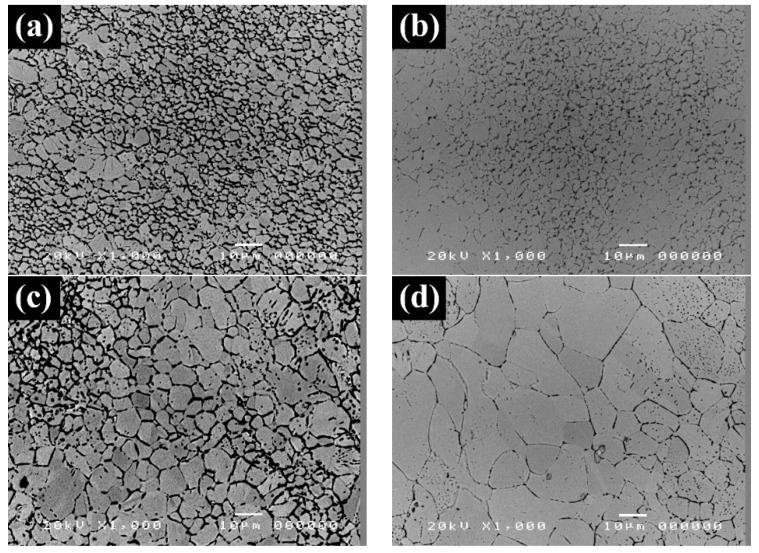
Microstructure of Al_0.3_CoCrCu_0.3_FeNi with cold rolling and annealing at 900 °C for (**a**) 120; (**b**) 300; (**c**) 600 and (**d**) 1200 min.

**Figure 7 entropy-20-00967-f007:**
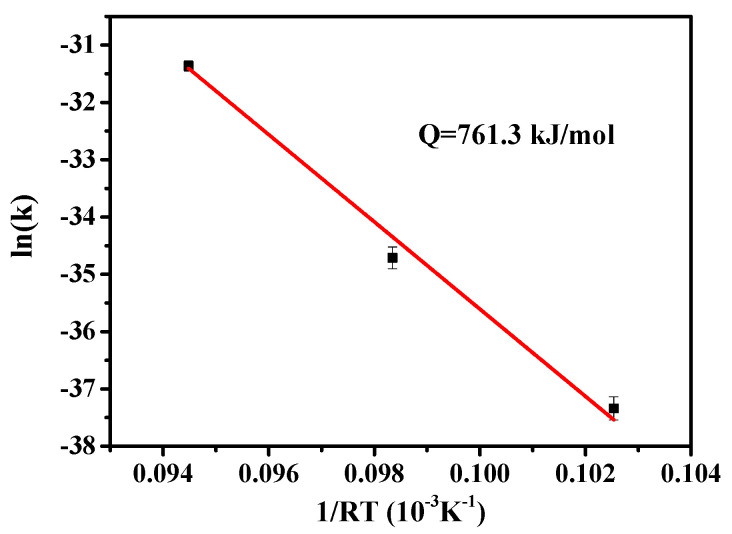
Grain growth activation energy of Al_0.3_CoCrCu_0.3_FeNi.

**Figure 8 entropy-20-00967-f008:**
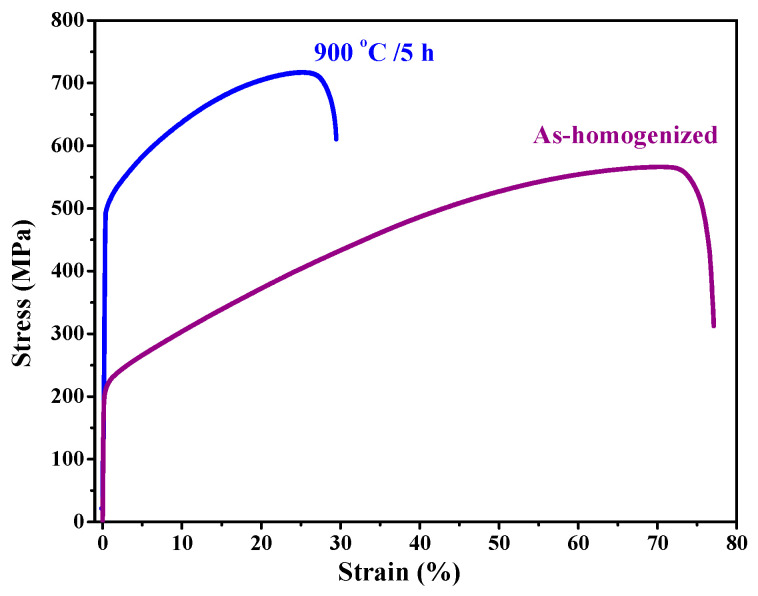
Mechanical properties of Al_0.3_CoCrCu_0.3_FeNi alloy annealed at 900 °C for 5 h and the as-homogenized state.

**Table 1 entropy-20-00967-t001:** Grain size of Al_0.2_CoCrFeMnNi annealed at 1000 °C, 1050 °C, and 1100 °C with different times (unit: μm).

Grain Size (μm)		Annealing Temperature (°C)		
		1000	1050	1100
Annealing time (min)	10 30 60 120	27.0 40.1 48.1 63.2	42.6 65.7 80.3 98.4	80.2 118.1 152.8 201.2

**Table 2 entropy-20-00967-t002:** Grain size of Al_0.3_CoCrCu_0.3_FeNi annealed at 900 °C, 950 °C, and 1000 °C at different times (unit: μm).

Grain Size (μm)		Annealing Temperature (°C)		
		900	950	1000
Annealing time (min)	120 300 600 1200	3.35 5.13 8.21 10.50	12.9 16.9 21.7 30.3	50.8 70.6 103.3 141.0

**Table 3 entropy-20-00967-t003:** Value of atomic size difference and grain growth activation energy.

	Atomic Size Difference (%)	Grain Growth Activation Energy (kJ/mol)	ΔH_mix_ (kJ/mol)
Al_0.3_CoCrCu_0.3_FeNi	3.65	761.3	−4.38
Al_0.2_CoCrFeMnNi	2.77	434.4	−6.24
CoCrFeMnNi	1.18	325.0	−4.16

**Table 4 entropy-20-00967-t004:**
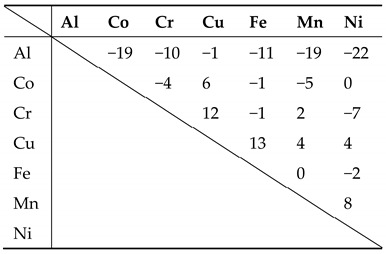
ΔH_mix_ for the element pairs (unit: kJ/mol).

**Table 5 entropy-20-00967-t005:** Comparisons of mechanical property between CoCrFeMnNi [6] and Al_0.3_CoCrCu_0.3_FeNi.

	CoCrFeMnNi [6]		Al_0.3_CoCrCu_0.3_FeNi	
Annealing condition temperature/time	1150 °C/1 h	800 °C/1 h	As-homogenized	900 °C/5 h
Grain size (μm)	155	4.4	516	5.13
YS (MPa)	190	350	217	500
UTS (MPa)	560	650	566	717
Elongation (%)	78	60	77	29

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
