# Peer review of "Effect of Atomic Size Difference on the Microstructure and Mechanical Properties of High-Entropy Alloys"

_entropy, 2018, doi:10.3390/e20120967_

Round 1

Reviewer 1 Report

Reviewer’s Comments on manuscript Entropy-388633 titled “Effect of Atomic Size Difference on the Microstructure and Mechanical Properties of High-3 entropy Alloys”.

The authors have presented results on change in activation-energy for grain-growth, on addition of solute elements like Al and Cu in equiatomic high entropy alloy compositions. Additionally, the authors have discussed the effect of solute addition on solid solution strengthen-abilty of HEA matrix (fcc matrix, in the alloys under consideration). The authors have compared their results to a well-published CoCrFeNiMn HEA. The experimental design is interesting; however, their conclusions and interpretations are pre-mature and incomplete.

1)      The authors compare CoCrFeNiMn with Al0.2CoCrFeNiMn where Al is added as the solute element. However, the third alloy Al0.3Cu0.3CoCrFeNi does not contain Mn, which makes it an un-systematic comparison.

2)      The authors need to do major corrections in the writing style as many phrases used are grammatically and technically incorrect. Eg.

Line 29-  High entropy causes microstructures to become stable; (probably authors meant single phase microstructures to be stable)

Line 33-  thereby hardening high-solid solutions in HEAs – needs correction

Lines 33-36- Lattice distortion impedes atomic movement and slows down the diffusion rate of atoms in HEAs; these conditions lead to high recrystallization temperatures, that is, the activation energies of grain growth in deformed HEAs are higher than those in conventional alloys [1,5].

The authors of the references cited did not provide any evidence of change in activation energy of grain growth or comparison with the conventional alloy!

Line 54- With a moderate amount of Al addition, a FCC single-phase structure and a high atomic size difference should be obtained in the designed alloy.

At what temperature and on what basis? No references.

Line 59- To design single-phase structures, researchers decreased the amounts of Al and Cu in Al0.5CoCrCuFeNi.

Which researchers, no example or reference is provided

3)      The authors did not talk about the value of n in equation 2. The value of n has a significance in terms of following a parabolic grain growth (n=2) or solute drag (n=3).

4)      The temperatures used for the grain growth for Al0.2CoCrFeNiMn are 1100, 1050 and 1000 and that for Al0.3Cu0.3CoCrFeNi are 900, 950 and 1000 C. It is well know that the temperature can change the mechanism of grain growth hence same temperatures should have been used to make a fair comparison.

5)      The authors report the lattice parameters of the alloys using XRD to the 3rd decimal place and the values are only changing on the 3rd place. What is the statistical accuracy or error margin of such results?

6)      Cu has a strong clustering tendency in these alloys due to positive heat of mixing. Especially at lower temperatures like 900 and 950 C the solubility of Cu can be as low as 1-2 % in this composition.

7)      The claims of lattice distortion resulting in this humongous change in activation energy need much more careful examination. The possibility of presence of cu-clusters within the fcc matrix cannot be completely discarded, that will only change the mechanism of grain growth but also influence the mechanical properties. Cu clusters have been used in many commercial steels as strengthener.

8)      Hence, either more high resolution details like TEM or APT examination of the alloys or thermodynamic assessment by CALPHAD of the compositions at the temperatures of experiments are needed to conclusively show that the alloys are indeed single phase. Without that, comparison of these three alloys is not a valid comparison as the mechanism of grain growth can be completely different in each alloy.  

Here are few examples (references) where Cu clustering has been clearly pointed out and utilized for strengthening in steels and HEA compositions.

Vaynman, Semyon, Dieter Isheim, R. Prakash Kolli, Shrikant P. Bhat, David N. Seidman, and Morris E. Fine. "High-strength low-carbon ferritic steel containing Cu-Fe-Ni-Al-Mn precipitates." Metallurgical and Materials Transactions A 39, no. 2 (2008): 363-373.

Jiao, Z. B., J. H. Luan, Z. W. Zhang, Michael K. Miller, W. B. Ma, and C. T. Liu. "Synergistic effects of Cu and Ni on nanoscale precipitation and mechanical properties of high-strength steels." Acta Materialia 61, no. 16 (2013): 5996-6005.

Wen, Y. R., A. Hirata, Z. W. Zhang, T. Fujita, C. T. Liu, J. H. Jiang, and M. W. Chen. "Microstructure characterization of Cu-rich nanoprecipitates in a Fe–2.5 Cu–1.5 Mn–4.0 Ni–1.0 Al multicomponent ferritic alloy." Acta Materialia 61, no. 6 (2013): 2133-2147.

Gwalani, B., D. Choudhuri, V. Soni, Y. Ren, M. Styles, J. Y. Hwang, S. J. Nam, H. Ryu, Soon Hyung Hong, and R. Banerjee. "Cu assisted stabilization and nucleation of L1 2 precipitates in Al 0.3 CuFeCrNi 2 fcc-based high entropy alloy." Acta Materialia 129 (2017): 170-182.

Ayyagari, Aditya V., Bharat Gwalani, Saideep Muskeri, Sundeep Mukherjee, and Rajarshi Banerjee. "Surface degradation mechanisms in precipitation-hardened high-entropy alloys." npj Materials Degradation 2, no. 1 (2018): 33.

Verma, A., P. Tarate, A. C. Abhyankar, M. R. Mohape, D. S. Gowtam, V. P. Deshmukh, and T. Shanmugasundaram. "High temperature wear in CoCrFeNiCux high entropy alloys: The role of Cu." Scripta Materialia 161 (2019): 28-31.

Author Response

We thank the reviewer for the positive comments and the helpful suggestions on how to improve our manuscript. A detailed description of our changes is in the attached file.

Reviewer 2 Report

   In this work, the authors have performed a set of experiments to investigate the effects of atomic size difference on the microstructure and mechanical properties of some HEAs, i.e. MnCoCrFeNi, Al0.2MnCoCrFeNi, Al0.3Cu0.3CoCrFeNi (all have single phase FCC structure). Their results indicate that the “activation energy” of grain growth (after recrystallization of cold rolled specimens) in these alloys increases by increasing the Al content from 0 to 0.3 molar ratio. Using a very simple model, i.e. Eq (1), they have postulated that increasing of the Al content leads to increasing of the atomic size difference, which in turn leads to increasing of the lattice distortion and hence, leads to higher activation emery of grain growth and “better” mechanical properties.

   While there is no considerable novelty in the findings and postulations of this manuscript, there are critical comments and questions which must be addressed so that studying the manuscript might be useful for the readers interested in HEAs. 

Major comments

1-      The concept of “lattice distortion” and its extent s is very interesting and arguable in HEAs. However, the authors have overlooked it!  

The authors have not provided any experimental evidence (or measurement) of difference between lattice distortion in their HEAs. There are numerous experimental/theoretical publications on definition and measurement of lattice distortion in HEAs which must be reviewed here [e.g. see” Owen and Jones, “Lattice distortions in high-entropy alloys”- J Mater Res 33, 2954 (2018), and references therein]. However, the authors of this manuscript have evaluated the lattice distortion merely based on the atomic size difference, δ, from Eq. (1). This is arguable because:

1-1-            It has been shown that the atomic size mismatch evaluated with the empirical atomic radii is not accurate enough to describe the local lattice distortion of HEAs. [H. Song et al., “Local lattice distortion in high-entropy alloys – Phys Rev Mater. 1, 023404 (2017)]

1-2-            Moreover, the atomic size of an element itself is not well defined and might be dependent on the chemical composition of the alloys [V.A. Lubarda, "On the effective lattice parameter of binary alloys" Mechabics of Materials 35, 53 (2003)].

2-      The selection of composition of HEAs is not systematic

To maintain the FCC structure of their base MnCoCrFeNi with Al molar ration of 0.3, the authors have completely replaced the 1 molar of Mn content with 0.3 molar ration of Cu in their Al0.3Cu0.3CoCrFeNi. Rather than atomic size variations, this replacement might have important effects on the chemical affinities and alloy properties (for example, according to Table 4, the mixing enthalpy of Al-Mn is -19 KJ/mol while that of Al-Cu is only -1 KJ/mol, etc.). Why the authors have not, for example, keep their base-HEA fixed and just increase the Al content, something like: AlxMnCoCrFeNi, where x=0, 0.1, 0.15, 0.2?

3-      Fig 8 and it corresponding discussions (lines 183 to 194) needs to be rewritten for the sake of clarification.

To this end, the authors have to discuss the following parameters and the corresponding variations with the composition of their HEAs:

3-1- What is the “driving force” for recrystallization?

3-2- What is the activation energy (energy barrier) of recrystallization?

3-3- What is the driving force for grain growth?

3-4- What is the energy barrier against of growth?

3-5- How does the activation energy of (sluggish) diffusion affect the grain growth?

And last but not least:

3-6- The energy of a defected lattice (e.g. a lattice with a dislocation) must be higher than the energy of the perfect lattice. Why in Figure 8, Egrain, undistorted is above (greater that) EPerfect_Dislocation (and similarly, for the case of distorted lattice in Fig 8)?

Minor comments

4-      In figure 2, the peak for (200) plane is disappeared in the water-quenched sample. In contrast, in Figure 3, the peak for (220) plane is emerged in the water-quenched sample. Such effects of different heat treatment on XRD patterns have not been discussed.

5-      In this manuscript, the claim of “better” mechanical properties are subjective and would be better to not overemphasized.

For example, according to Fig. 9, while the UTS of annealed sample annealed at 900 C is higher than that of homogeneous sample, its ductility is drastically lower. This recalls the well-known strength-ductility paradox (higher strength à lower ductility) in conventional alloys.

   5-1- Why the mechanical properties of Al0.2MnCoCrFeNi has not been given for better comparison with other two alloys?

6-      Eq. (1) needs to be modified (a power of 2 is missed under the square root):    … (1-ri/rave)2

7-      In the caption of Fig 3, the HEA is: Al0.3Cu0.3CoCrFeNi

Conclusions

Overall, this work does not have a considerable novelty, it suffers from lack of experimental evidence for the central claims, and the postulations are not strong and even are arguable.

Author Response

(The authors gave the same response as above.)

Reviewer 3 Report

The paper deals with the influence of the atomic sizes on the structural and mechanical characteristics of single face-centered cubic (FCC) phase high-entropy alloys. The paper is well written and the experimental results are interesting. I recommend its publication addressing the following comments:

-    Errors on the activation energy values of grain growth should be estimated by the linear fitting presented in Figures 5,7.

-     Errors on the mechanical parameters (Table 5) should be reported.

-   Exprimental details (energy of beam and working distance) for SEM analyses should be indicated.

-   Please report an additional figure containing the grain sizes distributions determined by the statistical analysis of SEM images.

Author Response

(The authors gave the same response as above.)

Round 2

Reviewer 1 Report

The authors have made no to very little change in the revised manuscript to answer the questions I raised in the previous round of review. Their conclusions are fundamentally incorrect.  Especially, the claims of lattice distortion resulting in this humongous change in activation energy need much more careful examination.  Their reply regarding the choice of alloys for the comparison is also unsatisfactory. 

I would not recommend the publication of the manuscript in Entropy.

Author Response

We thank the reviewer for the positive comments and the helpful suggestions on how to improve our manuscript. The attached file is detail description of our changes.

Reviewer 2 Report

Effect of Atomic Size Difference on the 2 Microstructure and Mechanical Properties of High-3 entropy Alloys

Journal: Entropy (id: 388633)

The authors have not properly used the opportunity for “major” revising of their manuscript by overlooking some critical comments, as the following: 

Response to Comment 1 & 2:

The authors’ have merely cited two introduced references, without carefully studying, comprehending, and discussing the essence of these references and the comments given. I further explain the issue in the following:

The authors have NOT provided any convincing experimental evaluation/measurement of “lattice distortion” in their systems, even though it has a key role in their discussions and conclusions. Indeed, they have ONLY used an arguable hypothetical formula, Equation 1, to evaluate the overall lattice distortion based on the differences in the atomic size of the alloys constitutive elements, which is even overemphasized by the title of their manuscript: “Effect of atomic size difference on the … “.

Changing the chemical composition of their alloy by replacing the Mn content with Cu (to preserve the FCC structure), may have important side effects due to considerable difference of chemical affinities of Cu vs. Mn. This fact is totally overlooked in the conclusions of current manuscript. For example, the increase of “activation energy” by changing the chemical composition can also be affected by the chemical bonding and diffusion/dislocation dynamics during grin growth. Hence, the extent of lattice distortion effect is not clear here. That’s why according to Ref [H. Song et al., “Local lattice distortion in high-entropy alloys – Phys Rev Mater. 1, 023404 (2017)], a precise/quantitative evaluation of the lattice distortion and its effect on properties of HEAs is yet an open question in high entropy alloys, and the major conclusion of this manuscript could be even misleading then. This concept must be carefully considered/discussed by the authors.

Response to Comment 3:

Figure 8 still has problems! In this figure, the authors have mixed up two concepts of “driving force” vs. “activation energy” for the grain growth phenomenon. As correctly discussed in lines 190 to 195 of their revised manuscript, the driving force of grain growth in HEAs could be smaller than that of low concentration alloys. Yet, the activation energy of grain growth in HEAs could be larger than that of low concentration alloys.

   Indeed, the energy levels showed in Fig. 8 could be used to illustrate the “driving force” of grain growth in HEAs vs. that in low concentration alloys. These energy levels cannot be used to illustrate the difference of “activation energies”, even though there might be some correlations.  (Please recall that the activation energy of sluggish diffusion also contributes in overall activation energy of grain growth in HEAs). So, lines 205 and 206 must be revised.  Here, I strongly suggest the authors to refer the Figure 3.16 of the book [Michael C. Gao, et al, "High-Entropy Alloys: Fundamentals and Applications", Springer (2016)] for better understanding of the concept of energy levels in different lattices and modifying their Figure 8. Here, is a link which would be helpful:

https://books.google.com/books?id=LEcWDAAAQBAJ&pg=PA83&lpg=PA83&dq=%22Diagram+shows+the+effect+of+Suzuki+interaction+and+lattice+distortion+on+stacking+fault%22&source=bl&ots=97dDOVuRRp&sig=7eJQS8UH9ITw1Prni0hu4FCiA5I&hl=en&sa=X&redir_esc=y#v=onepage&q=%22Diagram%20shows%20the%20effect%20of%20Suzuki%20interaction%20and%20lattice%20distortion%20on%20stacking%20fault%22&f=false

Authors note that:

1-    The energy level of a lattice with defects (e.g. a dislocation or stacking fault) must be higher (above) than the energy level of a perfect lattice.

2-    To modify their Fig 8, the authors can replace the caption of Ustacking_fault with UPerfect_Dislocation.

3-    The difference between these energy levels still indicates the “driving force” for grain growth, and NOT the “activation energies” (i.e. the energy barriers).

Conclusions

Overall, this work still does not have a considerable novelty, it suffers from lack of experimental evidence for the central claims, and the postulations are not strong and even are arguable.

Author Response

(The authors gave the same response as above.)

Reviewer 3 Report

The paper can be published in the present form.

Author Response

We thank the reviewer for the positive comments and the helpful suggestions on how to improve our manuscript. Your kind assistance will be much appreciated.